# Different Therapeutic Approaches for Dry and Wet AMD

**DOI:** 10.3390/ijms252313053

**Published:** 2024-12-04

**Authors:** Nicoletta Marchesi, Martina Capierri, Alessia Pascale, Annalisa Barbieri

**Affiliations:** Department of Drug Sciences, Section of Pharmacology, University of Pavia, 27100 Pavia, Italy; nicoletta.marchesi@unipv.it (N.M.); martina.capierri01@universitadipavia.it (M.C.); alessia.pascale@unipv.it (A.P.)

**Keywords:** wet AMD, dry AMD, neurodegenerative disease, anti-VEGF, dietary supplements

## Abstract

Age-related macular degeneration (AMD) is the most common cause of irreversible loss of central vision in elderly subjects, affecting men and women equally. It is a degenerative pathology that causes progressive damage to the macula, the central and most vital part of the retina. There are two forms of AMD depending on how the macula is damaged, dry AMD and wet or neovascular AMD. Dry AMD is the most common form; waste materials accumulate under the retina as old cells die, not being replaced. Wet AMD is less common, but can lead to vision loss much more quickly. Wet AMD is characterized by new abnormal blood vessels developing under the macula, where they do not normally grow. This frequently occurs in patients who already have dry AMD, as new blood vessels are developed to try to solve the problem. It is not known what causes AMD to develop; however, certain risk factors (i.e., age, smoking, genetic factors) can increase the risk of developing AMD. There are currently no treatments for dry AMD. There is evidence that not smoking, exercising regularly, eating nutritious food, and taking certain supplements can reduce the risk of acquiring AMD or slow its development. The main treatment for wet AMD is inhibitors of VEGF (vascular endothelial growth factor), a protein that stimulates the growth of new blood vessels. VEGF inhibitors can stop the growth of new blood vessels, preventing further damage to the macula and vision loss. In most patients, VEGF inhibitors can improve vision if macular degeneration is diagnosed early and treated accordingly. However, VEGF inhibitors cannot repair damage that has already occurred. Current AMD research is trying to find treatments for dry AMD and other options for wet AMD. This review provides a summary of the current evidence regarding the different treatments aimed at both forms of AMD with particular and greater attention to the dry form.

## 1. Introduction

Age-related macular degeneration (AMD) is the leading cause of irreversible central vision loss in elder adults. AMD is a progressive neurodegenerative disease which affects the macula, the central area of the retina responsible for sharp straight-ahead vision.

Nearly 200 million people worldwide are affected by some form of AMD. In the United States, the number is over 10 million and constantly rising. The prevalence of AMD varies by ethnicity: people of European descent have a higher risk than Asians and Africans. Hispanics are generally considered to be at low risk compared to Europeans, but the prevalence of AMD in Hispanics over the age of 80 is still over 8% [1]. A 2014 meta-analysis of population studies involving individuals aged 45–85 years estimated a prevalence of 8.69% for any form of age-related macular degeneration (AMD). This equates to 8.01% for early AMD and 0.37% for late AMD. Additionally, the same analysis revealed a prevalence of 0.44% for geographic atrophy and 0.46% for neovascular AMD [2].

The stages of AMD can be identified by clinical and imaging findings involving changes in the neural, structural, and vascular layers of the retina including photoreceptors, the retinal pigmented epithelium (RPE), the Bruch membrane, and the choriocapillaris [3]. Early and intermediate stages are diagnosed based on the presence of drusen—extracellular deposits of lipids, minerals, and proteins, also derived from apoptotic cells—and pigmentary changes, including both hypopigmentary and hyperpigmentary abnormalities. Specifically, early AMD is characterized by the presence of medium drusen (>63 μm and ≤125 μm) and the absence of pigmentary abnormalities, while intermediate AMD features larger drusen (>125 μm) and/or pigmentary abnormalities [4].

Visual loss associated with AMD typically manifests in the later stages of the disease as a deterioration of high-resolution central vision due to the interplay of various underlying processes. Dry and wet AMD represent the advanced forms of AMD [5].

Dry AMD, also referred to as geographic atrophy (GA), is characterized by areas of atrophy involving the RPE and the photoreceptor layer [3]. Photoreceptors are cells with an intense metabolism, and to function properly they need a high supply of oxygen. Photoreceptor damage is caused by photo-oxidation and free radical-induced lipid peroxidation. This could cause damage to the functions of the RPE and lead to damage to the macula. Products derived from the oxidative process are deposited in healthy tissue and cause slow and progressive cellular degeneration. The deposits hinder exchanges between blood vessels and the retina, and as a consequence the photoreceptors no longer receive enough nourishment, degrade, and cease functioning. In dry AMD, the eye is no longer able to completely dispose of metabolic waste, and the so-called drusen accumulate in the form of specks under the retina. The presence of these deposits hinders the supply of photoreceptors. A dry form can evolve into a wet one, which manifests itself as a response of the tissues to insufficient supply and lack of oxygen. Moreover, recent evidence suggests that metabolism in the retina is highly coordinated. For proper functioning of both cell types, the interaction of metabolism between the photoreceptors and RPE is essential, with RPE being responsible for the absorption of excessive light to reduce oxidative stress, the turnover of the external segments of the photoreceptors, glucose transport from the choroid to the photoreceptors, and the secretion of VEGF. So, mitochondrial damage in RPE is a trigger for degeneration in AMD [6].

Wet AMD is less common than GA; nonetheless, it is responsible for 90% of the severe central vision loss in patients affected by AMD. It is characterized by an abnormal expression of vascular endothelial growth factor (VEGF) which plays a key role in angiogenesis, promoting endothelial cell proliferation and migration, especially in the subretinal and outer retinal areas and the choroid. Furthermore, VEGF is responsible for an increase in the permeability of the blood vessels, meaning newly formed blood vessels have the tendency to leak fluid and cause hemorrhages and fibrosis [7].

The gold-standard therapy for patients with wet AMD is represented by the intravitreal injection of anti-VEGF molecules, able to limit VEGF function. With respect to the dry form, at present there is no effective therapy, other than intervention regarding the risk factors associated with lifestyle and diet. Herein, we summarize updates and new evidence concerning AMD treatments.

## 2. Risk Factors

AMD is a multifactorial disease whose onset and development are due to a complex interplay between a variety of factors including aging, genetic susceptibility, and environmental factors.

**Aging.** AMD is the main cause of vision impairment among adults over the age of ~55, and its prevalence is estimated to rise because of the aging of the population, reaching 288 million of affected individuals worldwide by 2040 [8]. The prevalence of AMD increases exponentially with age (OR 4.2 per decade). Rudnicka et al., in 2011, performed a systematic review and estimated that the prevalence of late AMD is 1.4% at 70 years of age, 5.6% at age 80 and 20% at age 90 [9]. Provided that, aging certainly plays a pivotal role in the development and progression of AMD. As individuals age, critical changes in the anatomy of the eye occur, which impact visual function and may contribute to the development and progression of neurodegenerative diseases. The most pronounced variation implies a decrease in choriocapillaris density in the RPE, Bruch membrane, and choroid. Therefore, blood supply to the RPE and Bruch membrane is diminished, as well as the clearance of lipoproteins [10]. As a consequence, cellular debris, along with deposits of lipoprotein-like and cholesterol-rich particles, which contribute to the infiltration of inflammatory mediators, accumulate in the Bruch membrane. This lipid layer is thought to be a precursor of AMD lesions [11]. Also, the RPE undergoes age-related structural changes. Cells shrink and become vacuolated, resulting in a general thinning of the RPE layer [12]. Moreover, melanin pigmentation diminishes over time; hence, its antioxidant abilities are reduced, also because of photobleaching of melanosomes [13]. Lipofuscin accumulates in the RPE over time as a result of the incomplete lysosomal digestion of phagocytosed photoreceptor outer segments. This molecule is proven to contribute to oxidative stress since it is involved in the production of free radicals and is responsible for the inhibition of the phagocytosis of damaged biomolecules and organelles [14].

**Genetic susceptibility.** Several studies show the contribution of genetic factors in AMD development [15,16,17,18]. The Beaver Dam Eye Study (BDES) is a population-based study which began in 1987, conducted on 4926 people between the ages of 43 and 86 years, the objective of which was to ascertain the prevalence, incidence, and potential risk factors involved in the development of AMD. Analysis of sibling correlations allowed researchers to underscore significant correlations between siblings and AMD, suggesting a genetic predisposition to the condition and the potential hereditary nature of the disease. Moreover, segregation studies suggested that a major gene effect could account for approximately 62% and 59% of the variance in age-related maculopathy scores for the right and left eyes, respectively [19]. These findings have paved the way for further molecular genetic investigations. One of the largest genome-wide association studies of AMD conducted by the International AMD Genomics Consortium (IAMDGC) identified 52 common and rare protein-altering variants distributed across 34 loci independently associated with AMD. These single nucleotide polymorphisms (SNPs) account for 27% of the risk for late AMD. A number of biological pathways, including inflammation and immunity, lipid metabolism and transport, cellular stress and toxicity, and extracellular matrix maintenance pathways, display significant enrichment, which suggests their potential role in AMD pathogenesis [20]. The two loci with the highest population attributable risk of developing AMD are Complement Factor H (*CFH*) and Human High-Temperature Requirement A-1/Age-Related Maculopathy Susceptibility 2 (*HTRA1/ARMS2*).

*CFH* on chromosome 1 is part of the Regulator of Complement Activation (*RCA*) gene cluster [21]. *CFH* plays a key role in the regulation of complement activation and represents a defense against lipid peroxidation and oxidative stress [22,23,24,25]. Therefore, *CFH* dysfunction may cause chronic inflammation, altered lipid metabolism and vulnerability toward oxidative stress. The Y402H polymorphism (rs1061170) leads to a substitution of tyrosine for histidine at position 402 and accounts for approximately 50% of AMD cases, particularly among Caucasians [26]. It is noteworthy that the Y402H variant is linked to diminished responsiveness to antioxidant therapy among individuals with non-neovascular AMD, while it seems to elicit a strong positive response to anti-VEGF treatments in patients with neovascular AMD [27]. Several rare variants of *CFH* have been identified, some of which are highly penetrant, such as R1210C, R53C, D90G, and P503A [28].

*ARMS2/HTRA1* is a gene located on chromosome 10 and its mutations strongly affect the progression of AMD. The rs10490924 polymorphism (A69S) in the ARMS2 gene is the most studied and consists of the substitution of amino acid 69 from alanine for serine. It has been demonstrated that the rrs10490924 polymorphism is more significant in Asian populations than in European populations, with a risk allele frequency of 40% and 20%, respectively [29]. The underlying mechanism is still not fully understood. It seems that *ARMS2/HTRA1* is implicated in the response against oxidative stress since its variants compromise the superoxide dismutase 2 response [30]. The *ARMS2* gene encodes a protein that may play a role in the complement system, potentially mediating the clearance of cellular debris resulting from apoptosis or necrosis. This function is crucial for maintaining retinal health [31]. In addition, *HTRA1* encodes a serine protease responsible for ECM remodeling and the inhibition of TGF-β, a key player in angiogenesis. For this reason, variations in *ARMS2/HTRA1* may lead, with a higher likelihood, to the development of wet AMD [32].

**Smoking.** Smoking is a major modifiable risk factor related to AMD onset and development. Compared to non-smokers, ex-smokers and current smokers are two- to four-fold more likely to develop AMD [33,34]. Several studies investigated whether the risk of AMD is correlated to the dose of cigarette consumption, and there is evidence that smoking significantly increases the risk of developing AMD in a dose-dependent manner [35,36,37,38,39]. Moreover, the risk persists up to 15–20 years after quitting smoking [36,40], and after that the likelihood of AMD is similar to never-smokers [41]. Smoking elicits detrimental effects on eye health; specifically, it may cause oxidative stress, lipid peroxidation, vascular changes, inflammation, and alteration of the RPE and Bruch membrane—all factors which may contribute to AMD progression [42,43]. Nicotine has vasculogenic properties that promote angiogenesis in experimental models by stimulating acetylcholine receptors (nAChRs) in endothelial progenitor cells [44]. Moreover, it causes nitric oxide (NO) secretion, which has vasodilating effects and pro-angiogenic properties. Nicotine promotes the release of catecholamines, responsible for platelet aggregability, contributing to the growth of plaque. Nicotine also exerts vasoconstrictive effects through α-adrenergic stimulation, which can impair blood flow in the choroid. Chronic exposure to cigarette smoke can lead to impaired normal angiogenic processes, contributing to AMD etiology. In addition, a metabolite derived from nicotine, nornicotine, has been proven to be responsible for impairing retinoid homeostasis, leading to the accumulation of lipofuscin in RPE cells and contributing to the growth of drusen and photoreceptor degeneration [34].

**Diet.** Dietary patterns and specific nutrients’ intake influence AMD progression, as proven by several population studies [42]. Adherence to the Mediterranean diet has been associated with a reduced risk of AMD. As proven in the population study performed by Barreto et al., individuals with high genetic risk who adhered to this diet experienced up to a 60% reduction in AMD risk compared to those with low adherence, who have a fourfold increase in AMD risk [45]. Conversely, an unbalanced diet rich in red and processed meats, fried foods, refined grains, and high-fat dairy products is associated with an increased risk of late-stage AMD [46]. The Mediterranean diet assumes a limited consumption of red meat and a substantial consumption of fruits, vegetables, whole grains, fish, and healthy fats, which contain high levels of vitamins, antioxidants, unsaturated fats, lutein, and zeaxanthin. The Age-Related Eye Disease Study (AREDS) is one of the most relevant large-scale clinical trials with the purpose of establishing the relationship between antioxidants intake and AMD progression. The AREDS2 supplementation formula contains lutein, zeaxanthin, vitamin C, vitamin E, Zn and Cu, and its consumption have been proven to reduce the risk of progressing to advanced AMD by 25% over a five-year period [47]. Furthermore, several cohort studies show evidence that regular intake of leafy greens, such as spinach, kale, and collard greens, is associated with a significant reduction in the likelihood of developing late AMD. These vegetables are recognized as rich sources of xanthophyll carotenoids, lutein, and zeaxanthin [48]. The latter nutrients have been proven to exert protective effects against oxidative stress, inflammation, and angiogenesis, the main players in AMD development [45]. Other important nutrients beneficial for AMD progression are fatty acids. Approximately 20% of the dry weight of the retina is composed of lipids, predominantly polyunsaturated fatty acids (PUFAs). These are particularly abundant in the outer segments of photoreceptors, where they enhance the fluidity of the photoreceptor membranes, facilitating a more rapid response to external stimuli [49]. Omega-3 (ω-3) is mostly present in fish, particularly in squid, cod, and tuna. There is evidence that ω-3 intake, specifically eicosapentaenoic acid (EPA), docosapentaenoic acid (DPA), and docosahexaenoic acid (DHA), may be beneficial against ischemia, and light, oxygen, and inflammatory damage contributing to AMD [50,51]. At present, dietary changes remain the primary therapeutic strategy to potentially delay or prevent the onset and progression of AMD.

**Comorbidities.** As we previously discussed and demonstrated with the literature, a correlation has emerged between AMD and particular lifestyle habits, but correlations also exist with other pathologies that somewhat depend on metabolism, such as diabetes, dyslipidemia, and hypertension. Some metabolic syndromes are considered as risk factors that accelerate aging in retinal neurons and may contribute to the neurodegeneration seen in age-related macular degeneration [52]. In particular, hyperglycemia and dyslipidemia occurring in diabetic patients are thought to disturb the homeostasis of the retina by inducing inflammatory responses in retinal tissue, including oxidative stress. It is suggested also that hyperthyroidism can accelerate the basal metabolic rate and oxidative metabolism by induction of mitochondrial enzymes, which may induce oxidative stress. Moreover, the possible involvement of hypertension in AMD is well defined: high blood pressure has been shown to be associated with lower choroidal blood flow and disturbed vascular homeostasis [53].

### Experimental Models for AMD

Experimental models for wet and dry AMD are summarized in Table 1.

The use of animal models of AMD has facilitated the acquisition of new insights into the underlying pathological mechanisms. The benefits of using rodent models in research include their cost-effectiveness, relatively rapid development times, and ease of genetic manipulation. Mouse models have been successful in recreating many of the histologic features of AMD, including the thickening of Bruch’s membrane, the development of drusen-like subretinal deposits, and immune dysregulation resulting in complement activation and the accumulation of macrophages or microglia.

Furthermore, laser-induced choroidal neovascularization (CNV) in rodents has constituted the foundation for the investigation of innovative treatments for wet AMD. The models presented in the table have facilitated the identification of the contributions of various pathological mechanisms, including genetic polymorphisms, oxidative damage, lipid and carbohydrate metabolism, and complement dysregulation.

Despite their utility, rodent models of AMD remain constrained by the anatomical absence of a macula. This limitation may explain why none of these models have yet been able to fully capture the complexity of AMD evolution. Nevertheless, they will continue to play a pivotal role in AMD modeling, even as new insights into its genetic and environmental underpinnings emerge [54].
ijms-25-13053-t001_Table 1Table 1In vitro and in vivo experimental models for wet AMD and dry AMD. RPE = retinal pigment epithelium; AMD = age related macular degeneration; CNV = choroidal neovascularization. (*) We summarized the main light-damage rodent models used for AMD, both wet and dry.
In VitroIn Vivo**Dry AMD**Defined models with structured experimental conditions and good reproducibility.**Primary RPE cells:** Human cells with natural differentiation.**ARPE cells:** Immortalized, rapid cell growth. Exhibit similar morphology and genetic makeup to primary RPE cells.**IPSC-RPE:** Exhibits similar morphology and genetic makeup to primary RPE cells.Some examples:Rabin et al.: Primary adult human RPE cells were isolated from cadaveric donor eyes. The subpopulation of RPE stem cells (RPESCs) was activated, expanded, and then differentiated into RPE progeny. Confluent cultures of RPESC-derived hRPE and ARPE-19 cells were exposed to a regimen of tert-butylhydroperoxide (TBHP) for 1–5 days [55]Maruotti et al.: hiPSC differentiation into RPE: high-throughput screening (HTS) analysis, using chronic exposure to A2E and blue light [56]Petroski et al.: Model with hallmark of dry AMD, pigment mottling of RPE, accumulation of intracellular lysosomal lipofuscin and extracellular drusen deposits [57]Virii and Marchesi et al.: model of autophagy and drusen [58,59]**Rodent**Animal models are important for determining which features need to be present to consider it a good model, i.e., the thickening of Bruch’s membrane (BM), sub-RPE basal laminar deposits and basal linear deposits (i.e., drusen), changes in the RPE, including loss of the basal infoldings, atrophy, and hyperplasia, accumulation of immune cells such as macrophages or microglia, deposition of activated complement proteins, photoreceptor atrophy, retinal or choroidal neovascularization, and fibrosis.Some examples:Complement factor pathway:Cfh^−/−^ mice [60]Transgenic CFH Y402H mice [61]Transgenic mice overexpressing C3 [62]C3a and C5a receptor^−/−^ mice [63]ChemokinesCcl2^−/−^ and Ccr2^−/−^ mice [64]Cx3cr1^−/−^ mice [65]Ccl2^−/−^ Cx3cr1^−/−^ double knockout mice [66]Oxidative damage models: animal models that lack intrinsic antioxidant mechanisms or those where additional oxidative stress is applied demonstrate many of the features of AMD.Immunization with carboxyethylpyrrole adducted proteins [67]Ceruloplasmin/hephaestin^−/−^ mice [68]Sod1^−/−^ mice [69]Cigarette smoke/hydroquinone^+/−^ high fat diet^+/−^ blue light [70]Nrf2/PGC-1 a double KO [71]Lipid/glucose metabolismAging mice^+/−^ high fat diet^+/−^ light treatment [72]APOE KO and transgenic mice [73,74]Light damage rodent models (*)Sprague Dawley rats: white light; 400–1000 lux [75,76]Wistar rats: white light and LED; 300–6000 lux [77,78]Long Evans rats: LED and constant fluorescence light; 500–6000 lux [79,80]C57BL/6J mice: white light, blue light; 800–100 k lux [81,82]BALB/c mice 4500–10,000 lux; white and blue light [83,84]ddY mice: 5000 lux; white fluorescent light [85]**2D models****Non-human primate models of dry AMD**Two-Dimensional models typically consist of a monolayer of cells, such as RPE cells, grown on flat surfaces.These models are easier to manipulate and analyze, making them suitable for initial drug screening and toxicological studies.They fail to replicate the complex interactions between different cell types and the extracellular matrix found in vivo. This can lead to oversimplified results that do not fully represent AMD pathology.Examples:−RPE cells derived from human embryonic stem cells or immortalized cell lines (e.g., ARPE-19) are often used in these models to study cellular responses to oxidative stress or other stimuli relevant to AMD [86,87]The retinal structure of non-human primates closely resembles that of humans, particularly in terms of the presence of a macula.Age-related maculopathy in rhesus and cynomolgus macaques [88]Early onset drusen in Japanese and cynomolgus macaques [89]**3D models**Three-dimensional models aim to mimic the layered structure of the retina more accurately. They can incorporate multiple cell types and extracellular matrix components, allowing for more physiologically relevant interactions.These models can better replicate the biochemical environment of AMD, including the formation of drusen and interactions between RPE and choroidal cells. They are particularly useful for studying disease mechanisms, drug responses, and potential therapeutic interventions. Some examples:−3D Outer Blood–Retinal Barrier (oBRB): Recent advancements have led to the development of a fully polarized RPE monolayer on top of a Bruch’s membrane with a fenestrated choriocapillaris network. This model successfully recapitulates both dry AMD phenotypes and wet AMD features, including choriocapillaris degeneration and sub-RPE drusen deposits [90,91]−Bioprinting Techniques: bioprinting to create vascularized retinal tissues that incorporate various cell types essential for studying AMD. This approach allows for precise control over tissue architecture and cellular composition [90,92]
**Wet AMD****Cell lines (as dry, with different treatment)****Murine Models**Defined models with structured experimental conditions and good reproducibility.**Primary RPE cells:** Human cells with natural differentiation.**ARPE cells:** Immortalized, rapid cell growth. Exhibits similar morphology and genetic makeup to primary RPE cells.**IPSC-RPE:** Exhibits similar morphology and genetic makeup to primary RPE cells.The replication of AMD through in vitro models has been the subject of many studies; for example:Golestaneh et al.: iPSC-RPE from patients with AMD versus healthy, treated with hydrogen peroxide [93]De Cilla et al.: used hydrogen peroxide to stress the cells, following treatments with aflibercept and ranibizumab [94]Wei et al.: human choroidal microvascular endothelial cells were incubated with VEGF and FGF [95]Kernet et al.: primary human RPE cells from healthy donors were illuminated under light exposure, that are able to induce cell death and, then be treated with sorafenib [96]Retinal degeneration genes in mice have a corresponding gene in humans, with many human gene orthologs existing in the mice genome. They have short lifespans, which enables us to see the aging process. Protocols for genetic studies are well established.**CNV**CNV replicates the neovascularization in neovascular AMD, is low cost, and it develops rapidly.The Ccl2/Ccr2 knockout model is used when studying nAMD, in which CNV can be easily induced [65]CXCR5 knockout mice present AMD symptoms such as Bruch’s membrane thickening, amyloid-β accumulation, RPE atrophy, and spontaneous neovascularization and drusen [97,98]
**Cocultures****Zebrafish Models**AMD is a multifactorial disease, which involves changes in RPE cells, in Bruch’s membrane, and the choroid. To accurately reproduce this complexity in vitro, a few studies were carried out on creation cell cocultures or cultivation cells on an artificial Bruch’s membrane. This enables cell-to-cell interactions and cross-talks, and modulation of cytokine production. More cytokines are usually produced in cocultures. Some examples:De Cilla et al.: coculture of RPE cells with HUVEC cells elicited a cross-talk between the cells when treated with aflibercept or ranibizumab [94]Jeong et al.: coculture of RPE and endothelial cells (HUVEC), carried out to study the modulation the production of TGF-β2 and VEGFR2 expression [99]Leclaire et al.: microglia (BV-2 cell line and primary cells from murine brain) in vitro with lipofuscin isolated from the human RPE to study inflammation [100]Ma et al.: primary RPE cells cocultured with microglia cells activated by lipopolysaccharide produced increased levels of pro-inflammatory cytokines [101]Zebrafish are a good recipient for treatments, as the drugs of choice can be added to the culture medium rather than injected into the fish, and they have high reproduction rates and transparent bodies (for close monitoring).They are ideal for the testing of anti-VGFR drugs, both by determining the cytotoxicity of the drug and its effect on angiogenesis.Wei et al. showed that lenvatinib suppressed angiogenesis of subintestinal vessels in zebrafish [95]Li et al. showed the same effects in embryonic zebrafish angiogenesis using brivanib [102]Hanovice et al.: showed that a reliable model of neovascularization could be induced in *fli:egfp zebrafish* (endothelial and lymphatic vessel expressing GFP) for hypoxia [103]
**2D models****Non-human primate models of wet macular degeneration**Collagen I and IV, fibronectin, matrigel, and gelatin are the most common biological substrates used in RPE cultures to mimic Bruch’s membrane. Examples:Murphy et al. used non-mammalian material like silk fibroin and alginate [104]Malek et al. created a model to mimic the disruption of the clearance of material in vivo and the accumulation of drusen [105]Altman et al. used silk fibroin from Bombyx mori for a scaffold [106]**CNV**First model: laser injury to disrupt Bruch’s membrane and induce CNV in a non-human primate, specifically the stumptailed macaque (*Macaca speciosa*) [107]In monkeys and rabbits [108]Laser-induced choroidal neovascularization in African green monkeys [109]
**3D models**Xiang et al. created a structure by combining electrospun PCL (polycaprolactone, a biodegradable polyester) nanotubes with silk fibroin and gelatin for ARPE-19 cells cultivation [110]Three-dimensional vascularized eye tissue models age-related macular degeneration [90].A 3D model, such as on involving electrospun fibers, is generally more relevant and should be favored when possible. Three dimensional models are more representative, both by the morphological aspects of cells and by the exposure and drug sensitivity [111]

## 3. New Treatments for Wet AMD

### Anti-VEGF Drugs

Wet AMD develops when new, abnormal blood vessels grow under the retina and leak blood or other fluids. The progression of vision loss is more rapid in individuals affected by wet AMD than those with dry AMD. In the early 2000s, scientists began developing drugs that interfere with this process. Specifically, they block a protein called vascular endothelial growth factor (VEGF), responsible for pathological angiogenesis. Prior to the development of these drugs, which are known as anti-VEGF drugs, individuals with wet AMD were almost certain to contract severe vision loss or blindness.

The gold-standard treatment proposed for patients with wet AMD is based on limiting the function of VEGF via intravitreal injection of anti-VEGF molecules, which were found to be effective in avoiding the loss of visual function and stabilizing disease progression [1,112]. All the anti-VEGF approved drugs are summarized in Table 2. Briefly, Ranibizumab^®^ is a humanized monoclonal antibody fragment which targets VEGF-A, Brolucizumab^®^ is a single chain fragment of humanized anti-VEGF-A antibody, and Aflibercept^®^ is a soluble protein which targets VEGF receptor (VEGFR). All drugs have shown protective effects in AMD patients, although recent studies demonstrate that Brolucizumab^®^ has better safety and efficacy profiles than Aflibercept^®^ [113], attributed to Brolucizumab^®^’s ability to cross retinal layers, leading to higher exposure in the retina and RPE/choroid [114]. On the other hand, recent studies report that a Ranibizumab^®^ 0.5 mg treatment (clinical dosage in use) for 12 months improved visual outcomes in the observational RENOWNED study [115], with the drug being the one with the lowest dose required for the treatment. Conbercept^®^ is approved for use only in China and binds to PLGF, VEGF-A, and VEGF-B, and VEGF-C. Despite the wide recommendation and clinical use of anti-VEGF treatment for AMD patients, the main reported disadvantage associated with this therapy is related to the short half-life of anti-VEGF molecules, and the consequent demand for repeated injections [116,117].

Overall, anti-VEGF drugs have demonstrated their successes, and we have a variety of anti-VEGF antibody-derived drugs. Additionally, patients have been able to benefit from anti-VEGF treatment for the last 20 years in the treatment of wet AMD. Several new candidates for anti-VEGF agents are under clinical trial, signifying a future for anti-VEGF drugs. For one, the recent bispecific antibody drug, Faricimab^®^ is a monoclonal antibody approved in 2022 against angiopoietin-2 and VEGF-A; this inhibitor shows promising results in extended time intervals between ocular injections (Table 2). This approach can control and inhibit molecules in different pathways associated with pathological conditions. Always in this way, a specific fusion protein called Efdamrofusp alpha (IBI302; Innovent Biologics, Shanghai, China) that was designed to and is thought to recognize both VEGF and C3b/C4b, and IBI302, is under phase 2 clinical trials (ClinicalTrials.gov IDs: NCT04820452, NCT05403749; https://clinicaltrials.gov/study/NCT04820452, accessed on 4 October 2024; https://clinicaltrials.gov/study/NCT05403749, accessed on 4 October 2024). The VEGF inhibitor domain of IBI302 is similar to that of Aflibercept, and the C3b/C4b inhibitor domain is derived from complement receptor 1.

Another approach implies improving the delivery system of anti-VEGF drugs in order to extend their duration. Susvimo^®^ was approved in 2021 and consists of a silicon shell containing a drug reservoir that is implanted into the eye through a sclera and pars plana incision and continuously releases Ranibizumab over a period of 6 months in the implanted eye. At the phase 3 clinical trial, Susvimo^®^ was non-inferior to Ranibizumab injection (0.5 mg every month) in gaining visual acuity within 9 months. However, Susvimo^®^ prescribing information warns of a threefold higher rate of endophthalmitis than the monthly IVT of Ranibizumab, as well as conjunctival erosion and retraction [118,119,120].

Additionally, targeted protein degradation technologies hold the potential for future applications. Bi-AbCap by Eli Lilly & Co (Indianapolis, IN, USA) is the first translational study for extracellular VEGF degradation [121] and consists of a fusion antibody system which is formed by an anti-insulin growth factor type I receptor (IGF-IR) antibody and the domain 2 of VEGFR1. The anti-IGF-IR and anti-VEGF bi-AbCap were designed to capture, internalize, and degrade extracellular VEGF using intracellular lysosomes. This targeted degradation technique was demonstrated in a cancer cell model, but it is applicable to other diseases, including wet AMD. This degradation strategy has been mostly limited to cytoplasmic proteins and many other studies are necessary to completely understand the power and mechanism.

Anti-VEGF injections have been shown in clinical trials to help over 90% of patients maintain their degree of vision; in actual practice, however, the number is closer to 50%. This is because people are not receiving the appropriate level of care on a regular basis. The issue is that most patients require an injection once every two months to maintain their vision. For many older patients who are dependent on others to transport them to their ophthalmology appointments and are dealing with other medical conditions, this can be a challenging schedule to follow.

Nowadays, a lot of the most fascinating research is investigating alternatives to routine injections. It is hoped that more constant therapy may help patients maintain more of their vision in addition to convenience.

Subretinal fibrosis is a significant complication associated with wet AMD. This condition can lead to severe visual impairment and is characterized by the presence of fibrotic tissue in the subretinal space, often following the formation of choroidal neovascularization (CNV). Literature data indicates a high cumulative incidence of subretinal fibrosis in patients with wet AMD. A recent study found that after 10 years, approximately 62.7% of patients developed fibrosis, with rates increasing significantly over time: 36% after one year, 45% after two years, and reaching up to 71% by the tenth year [122]. The presence of subretinal fibrosis is associated with poorer long-term visual outcomes, even among those receiving anti-VEGF treatments, which are standard for managing nAMD. Specifically, patients with subretinal fibrosis may experience a decline in best corrected visual acuity (BCVA) of more than 15 letters on the ETDRS scale over a decade. The formation of subretinal fibrosis occurs as a response to retinal injury and inflammation. Following CNV, inflammatory signals attract various cells, including fibroblasts, which transform into myofibroblasts and produce excessive extracellular matrix (ECM) components. This process leads to the development of fibrotic scars that disrupt normal retinal architecture and function [123]. Fibrotic tissue can significantly impair the communication between photoreceptors and the retinal pigment epithelium (RPE), leading to degeneration of photoreceptors and further visual loss. Recently, Armendariz and colleagues reviewed all the clinical trials in phase 2 and 3 for treatment, but as of yet there is no approved treatment. Several companies have tried to address this, without any success up to now. To develop an effective and safe treatment, it is necessary to have a complete understanding of the molecular pathophysiology and the role of the different cells involved, because fibrosis can contribute to resistance against anti-VEGF therapies, complicating treatment regimens and outcomes [124]. Between clinical and predictive factors, it is possible to account for the baseline visual acuity and the type of macular neovascularization that are predictive of fibrosis development. Patients with worse baseline BCVA are at higher risk for developing subretinal fibrosis [125].

## 4. Gene Therapy for Wet AMD

Gene therapy is a promising alternative to ongoing eye injections of drugs described above. The goal of gene therapy is to provide a “one-and-done” treatment by helping the eye with making its own anti-VEGF medicine. Two different methods are under investigation. One injects the gene therapy medium underneath the retina in a surgical procedure; the other injects it into the eye just like a routine anti-VEGF treatment, and is carried out in the ophthalmologist’s office. Despite the promise of gene therapy, the long-term effectiveness remains uncertain. Such a treatment is likely to be very expensive and may not be suitable for everyone with wet AMD.

On ClinicalTrials.gov 24 studies on gene therapy in wet AMD are listed (https://clinicaltrials.gov/search?cond=Wet%20AMD&intr=gene%20therapy, accessed on 9 August 2024). These include one study completed with results, five completed, thirteen recruiting, three active and not recruiting, one not yet recruiting, and one enrolling by invitation. In this section, we provide some brief details on two clinical trials, ADVM-022 and RGX-314, as they present the most advanced stages of such studies. The aim of these studies is to ensure a long-term expression of therapeutic levels of intraocular protein, allow for maintaining vision, and safely reduce the current treatment burden.

ADVM-022 has opened a revolutionary perspective in wet AMD treatment because it represents the first, and a recent, success (OPTIC trial, January 2024) of intravitreal gene therapy. The main aim of ADVM-022 was to treat wet AMD with a single intravitreal injection of the AAV2.7m8 capsid containing cDNA for an Aflibercept-like protein. In this way, retinal cells are able to continuously produce Aflibercept, a VEGF antagonist, after a single injection [126].

ABBV-RGX-314, developed in collaboration with AbbVie, is a promising gene therapy designed for the treatment of wet AMD. This therapy utilizes an adeno-associated virus (AAV) vector to deliver a gene encoding a monoclonal antibody fragment able to inhibit VEGF. ABBV-RGX-314 is currently undergoing evaluation in pivotal trials, including ATMOSPHERE and ASCENT™, which aim to determine its efficacy and safety as a one-time treatment option for wet AMD (ClinicalTrials.gov IDs: NCT05407636 https://clinicaltrials.gov/study/NCT05407636, accessed on 4 October 2024). Initial results from a Phase I/IIa trial published in *The Lancet* indicate that a single administration of ABBV-RGX-314 is generally well tolerated and can lead to stable or improved visual acuity and retinal thickness in patients, with few requiring supplemental anti-VEGF injections over a two-year follow-up period. The trial involved 42 participants who received a subretinal injection of the gene therapy vector, and the results showed sustained levels of the therapeutic protein in the aqueous humor, suggesting long-term efficacy. Most patients demonstrated stable or improved vision, highlighting the potential of ABBV-RGX-314 to reduce the treatment burden associated with frequent anti-VEGF injections that are typically required for AMD management [127,128,129].

The promising data from these trials have led to plans for global regulatory submissions to the U.S. Food and Drug Administration (FDA) and the European Medicines Agency (EMA) anticipated between late 2025 and early 2026. If successful, ABBV-RGX-314 could establish a new standard of care for wet AMD, significantly impacting the treatment landscape for this chronic condition.

## 5. Photodynamic Therapy

Wet AMD is possible to treat with regular eye injections (as described above) and, occasionally, with photodynamic therapy (PDT). It is recommended to associate PDT with eye injections if the latter alone does not help, which usually needs to be repeated every few months.

PDT involves the use of a photosensitizer, a dye capable of transforming light into chemical energy. This process produces singlet oxygen, a highly reactive species which induces cytotoxicity through the disruption of lipid membranes, proteins, and nucleic acids. In this way, PDT is able to cause site-specific vascular closure and cellular destruction. The damage to nearby tissues is minimal since the produced free radicals can affect molecules within a radius of about 100 nm only [130].

The first application of PDT in ophthalmology took place in the 1990s [131] and involves the intravenous infusion of Verteporfin as the photosensitive dye. Verteporfin was approved in 2002 by the FDA, clinically indicated for the treatment of choroidal neovascularization (CNV) in neovascular AMD, pathologic myopia, and ocular histoplasmosis [132]. In order to activate Verteporfin, an infrared laser is used, leading to the formation of singlet oxygen and free radicals that interact with endothelial cell membranes on choroidal blood vessels [133]. As a consequence, local immune-modulating factors like histamines, thromboxane, and TNF-α increase, causing vasoconstriction, thrombosis, increased vascular permeability, blood stasis, and hypoxia, promoting vaso-occlusion of pathologic blood vessels [134]. The most frequent adverse reaction of PDT is photosensitivity induced by Verteporfin [135].

In wet AMD, PDT represented a first-line treatment in 1999 after the Treatment of AMD with Photodynamic Therapy (TAP) study. The TAP trials enrolled 402 patients with CNV in Europe and in the US. Results show that patients treated with PDT had a higher percentage of eyes retaining baseline vision compared to the placebo (12 months: 61% treated, 46% placebo; 24 months: 53% treated, 37% placebo) [136]. In the 3-year TAP follow-up study, individuals treated with PDT demonstrated stable vision and no significant systemic safety problems over 5 years [137].

The ANCHOR study is a multicenter, international, double-masked, randomized control trial which, in 2009, enrolled 423 patients. This study highlighted that anti-VEGF therapy was more effective in the treatment of wet AMD compared to PDT. After 2 years, 90.0% of the patients who were treated with monthly Ranibizumab lost less than 15 letters of visual acuity in comparison to 65.7% of patients treated with PDT [138].

With the aim of investigating the efficacy of PDT in combination with anti-VEGF therapy, in comparison to anti-VEGF therapy alone, the MONT BLANC study enrolled 255 patients. The obtained data showed that at 1 year the combination group of Ranibizumab and Verteporfin PDT was non-inferior to Ranibizumab monotherapy [139].

Recently, it has been reported that PDT may lead to compensatory upregulation of angiogenic factors such as VEGF that may compromise its long-term efficacy against wet AMD. In this way, the combination of anti-VEGF therapy and PDT would be beneficial for the long-term treatment of wet AMD [140]. Xu and colleagues developed a nanosystem (Di-DAS-VER NPs) comprising reactive oxygen species-sensitive Dasatinib (DAS) prodrug and photosensitizer Verteporfin, administrable intravenously. Red light irradiation in the diseased eyes triggers the activation of the nanosystem, causing ROS production and intraocular DAS release, leading to vascular occlusion and CNV suppression. Systemic toxicity is not significant because the prodrug needs to be exposed to red light to be activated. The findings of this study may provide a promising clinical solution for minimally invasive administration of therapeutic agents for angiogenic eye diseases [141].

## 6. Dietary Supplements in Wet AMD

Currently, no pharmacological treatment exists to cure the more prevalent dry AMD form, while wet AMD is treated with intravitreal anti-vascular endothelial growth factor (anti-VEGF) inhibitors (as previously mentioned). Repeated injections and high drug costs represent a heavy load on ophthalmology clinics, and the frequent drug administrations are not patient friendly. Therefore, there is an urgent need for new treatment options. Dietary supplementation would represent a non-invasive and cost-effective option which would be expected to be highly patient-friendly and thus associated with good compliance. Several investigators have reported improvements in AMD progression using different supplement approaches. Furthermore, the Age-Related Eye Disease Study (AREDS) conducted between 1992 and 2001 showed that dietary supplements containing high levels of antioxidants and zinc could delay the progression of intermediate AMD to late AMD and vision loss (25% and 19% risk reduction as compared to placebo, respectively) [142]. From 2006 to 2012, AREDS was followed by AREDS2, which enrolled only patients with intermediate AMD [143]. AREDS2 was a very large research study. It looked at taking vitamins and minerals daily for AMD. These supplements may also lower the risk of wet AMD and vision loss in the second eye of people who lost vision in one eye from AMD. Taking the following nutritional supplements every day may help these people lower their risk of acquiring late-stage or wet AMD: Vitamin C (ascorbic acid) 500 mg, Vitamin E 400 international units (IU), Lutein 10 mg, Zeaxanthin 2 mg, Zinc (as zinc oxide) 80 mg and Copper (as cupric oxide) 2 mg. The outcomes of the two AREDS trials demonstrate that there is a clear potential for developing novel nutritional supplement formulations for delaying the progression of different AMD stages.

It is important to remember that nutritional supplements are not a cure for AMD but may help to slow the disease in some people with certain forms of AMD and, in particular, for wet AMD, may help to change the amount of needed anti-VEGF injections.

Dietary supplements have been investigated for their potential to influence the treatment and management of wet age-related macular degeneration (AMD), particularly in conjunction with anti-VEGF (vascular endothelial growth factor) injections. Recent studies suggest that certain supplements may alter the effectiveness and frequency of these injections.

A study demonstrated that omega-3 supplementation, when combined with anti-VEGF therapy, resulted in significantly lower levels of vitreal VEGF-A in patients with wet AMD. Specifically, patients receiving omega-3s showed vitreal VEGF-A levels of 141.11 pg/mL compared to 626.09 pg/mL in those not receiving omega-3s, indicating a potential reduction in the need for anti-VEGF injections due to decreased VEGF levels [144]. This suggests that omega-3 fatty acids may enhance the efficacy of anti-VEGF treatments, potentially leading to fewer injections over time.

Zeaxanthin supplementation is important in wet AMD. In particular, Zeaxanthin has been studied in conjunction with anti-VEGF treatments for conditions like nAMD. The effects of Zeaxanthin supplementation are different from reductions in disease progression in terms of reducing the cost of treatments.

In depth, regarding reduction in disease progression, a clinical trial indicated that oral zeaxanthin supplementation significantly decreased the conversion of atrophic fellow eyes to nAMD in patients undergoing anti-VEGF therapy. Specifically, the incidence dropped from 48% to 22%, marking a 54% reduction over five years [145].

While some studies have shown improvements in visual acuity with zeaxanthin supplementation, the latest findings suggest that there may not be a notable difference in vision between groups receiving anti-VEGF treatment alone and those receiving it alongside zeaxanthin (ClinicalTrials.gov ID: NCT01527435; https://clinicaltrials.gov/study/NCT01527435, accessed on 4 October 2024). However, the overall vision improvement in patients with unilateral wet AMD treated with triple therapy (anti-VEGF, corticosteroids, and photodynamic therapy) was observed, indicating a potential adjunctive role for zeaxanthin: a 2-year clinical trial [146] did demonstrate a risk reduction for AMD occurrence with 20 mg Zx supplementation versus a cohort control not receiving Zx supplementation when both took daily AREDS2 supplements containing 2 mg of Zx.

Keegan and coworkers, in 2020, showed that zeaxanthin, along with lutein, can attenuate VEGF-induced neovascularization in vitro. These carotenoids may reduce oxidative stress associated with VEGF, which is a key factor in the progression of AMD and other retinal diseases [147].

Overall, dietary supplements, particularly omega-3 fatty acids and carotenoids like zeaxanthin, may play a significant role in the management of wet AMD by potentially reducing the need for anti-VEGF injections. Ongoing research continues to explore these relationships, aiming to optimize treatment strategies for patients with AMD.

## 7. Treatments for Dry AMD

Currently there are no established treatments for dry AMD. In order to reduce the risk of AMD or slow its development, some evidence suggests that healthy and balanced nutrition, some supplements, such as vitamins or minerals, avoiding smoking cigarettes, and regular physical activity may be useful [3,148]. These strategies are mainly aimed at counteracting risk factors and providing a valid non-pharmacological treatment based on lifestyle changes. One useful form of management concerns the use of nutraceuticals, since the protective role of antioxidant food supplements against dry AMD is well known. These formulations are associated with an increased probability of reducing the rate of progression of dry AMD, although the long-term results are not as satisfactory as expected. In age-related maculopathy, the antioxidants used can belong to different families: minerals such as zinc, iron, and copper, carotenoids, bioflavonoids, vitamins (A, C, E, group B), and polyunsaturated fatty acids. It is well known that zinc and iron are elements that help with the eyes’ health. In particular, it has been discovered that zinc, being found in the retina, helps the functioning of the enzymes responsible for eye health. People who have maculopathy may suffer from low levels of zinc in the retina, so eating foods rich in zinc can definitely help. The most exploited antioxidant food belongs to the group of carotenoids, in particular to xanthophylls, which, unlike carotenes (beta-carotene and lycopene), are widely present in the human macula. The most important xanthophyll are lutein and zeaxanthin, present naturally in various foods, especially in vegetables [149,150].

Also worthy of attention is the spice saffron (*Crocus sativus* L.). Increasing evidence attributes several therapeutic properties to it, such as protection against ischemia, and anticonvulsant, antidepressant, anxiolytic, hypolipidemic, anti-atherogenic, anti-hypertensive, antidiabetic, and anti-cancer properties. In addition, several studies underline the antioxidant, anti-inflammatory, and antiapoptic properties of this compound, able to reduce oxidative damage and to protect cells from apoptosis. Based on this knowledge, saffron may protect photoreceptors, preserving retinal function in the initial stages of the disease, slowing down the progression of the neurodegenerative process of the retina, potentially due to the high concentration of carotenoids, mainly crocin and crocetin [151,152,153,154,155,156,157].

Several studies have revealed that regular consumption of foods containing ω-3 fatty acids is associated with a lower risk of AMD; thus, the Mediterranean diet, which contains foods rich in ω-3 fatty acids (walnuts, olive oil, and fish), appears to be protective against the progression of AMD. Furthermore, the association of the antioxidant properties of intestinal bacteria, such as Bacteroides and Eysipelotrichi, is associated with a lower risk of AMD in individuals whose microbiota are enriched [158].

Nevertheless, several therapeutic approaches are investigated as being able to reduce the rate of disease progression. Attempts are being made to attack dry AMD based on various aspects such as oxidative damage, chronic inflammation, the accumulation of toxins deriving from normal visual function. The approaches involve a wide range of targets and include drugs with antioxidative properties, inhibitors of the complement cascade, neuroprotective agents, visual cycle inhibitors, and gene- and cell-based therapies.

The encouraging data are supported by the availability of recent technologies, spectral OTC and autofluorescence, and non-invasive diagnostic methods, which allow for the evaluation of the possible positive effects of pharmacological treatments at the level of the photoreceptors and the retinal pigmented epithelium (RPE).

Some antioxidant molecules are available, added to the natural antioxidants introduced with the diet or supplemental food. The efficacy of OT-551 for the treatment of geographic atrophy was studied [159]. OT-551 is a disubstituted hydroxylamine with antioxidant properties as well as anti-inflammatory and antiangiogenic; it is administered as an eye drop, and the obtained results indicate a possible effect in maintaining visual acuity.

At the retinal level, mitochondrial dysfunction is the basis of oxidative stress, and it is induced by various negative factors such as cigarette smoking, accumulation of lipofuscin, and complement dysregulation. Elamipretide is a tetrapeptide believed to target cardiolipin in mitochondria, a lipid involved in energy transduction, thereby reducing reactive oxygen species production (ClinicalTrials.gov ID: NCT02848313; https://clinicaltrials.gov/study/NCT02848313, accessed on 4 October 2024). Administered in patients with dry AMD, it showed a possible positive effect on visual function. Risuteganib is an anti-integrin small peptide molecule that downregulates oxidative stress [160,161], targeting four different integrin heterodimers involved in the pathogenesis of AMD. Preclinical and early clinical studies show improvement in visual function, and it could slow down the progression of AMD.

Complement activation plays a key role in many chronic inflammatory processes such as those involved in the development of AMD. In fact, excessive activation of the complement cascade causes destruction of healthy cells, which may lead to the onset or progression of ocular pathologies. Complement activation stimulates surrounding RPE cells to secrete a range of inflammatory factors, such as monocyte chemotactic protein 1, interleukin 6, and interleukin 8, inducing chronic inflammation, which is a typical ocular change in AMD. Therefore, targeting the phases of this cascade allows for the control of degeneration. Pegcetacoplan, also known as APL-2, is one of the molecules that has been studied in recent years; it inhibits the cleavage of factor C3 into C3a and C3b [162], key steps in the complement cascade. C3 represents an advantageous therapeutic target, precisely because its effects span across all complement activation pathways. Moreover, POT-4 is a peptide capable of binding to complement factor C3, producing a potent inhibition of the complement activation cascade. The drug could find utility in dry forms of macular degeneration [163], since preliminary results have shown that intravitreal POT-4 is safe and well tolerated. In this context, due to its ability to inhibit the production and activity of inflammatory factors, the use of Sirolimus in AMD patients has been studied [164]. Contrary to expectations, Sirolimus did not result as being efficacious in this pathological condition. Immunosuppression may be important for some stages of the AMD process, but it is not the main mechanism involved.

In dry macular degeneration, lipofuscin, one of the waste products of photoreceptor metabolism, accumulates and leads to atrophy of pigmented epithelium cells, located behind the retina, and of the retinal photoreceptors [58]. Unwanted protein waste also aggregates outside the cellular environment and appears as yellow–white aggregates called “drusen”. These accumulations suffocate the retinal cells until they die. From there, the accumulation of lipofuscin in the retinal pigment epithelium is a hallmark of aging in the eye. In particular, lipofuscin causes numerous damages to cellular functions, inhibiting lysosomal degradation, and is photoreactive. In fact, it generates reactive oxygen species (ROS), proving to be phototoxic also for RPE cells. Moreover, it is also able to inhibit the action of the catalase enzyme (detoxifying, reduction action in ROS), sensitizing the cells and thus increasing the extent of damage [165]. The bis-retinoid N-retinyl-N-retinylidene ethanolamine (A2E), formed from retinal, has been identified as a byproduct of the visual cycle, and numerous in vitro studies have found toxicity associated with this compound [165].

Some drugs can modulate the visual cycle, slowing down the function of rods in particular, and thus can reduce the accumulation of toxic compounds in the retina. For this reason, pharmacotherapy with visual cycle modulators is under investigation for AMD.

Fenretinide is a synthetic derivative of vitamin A and its characteristics have been studied in regard to numerous diseases for its chemoprotective, apoptotic, anti-inflammatory, and antiangiogenic properties. This compound is a synthetic retinoid that blocks the process that leads to the formation of lipofuscin and its main toxic component A2E in the retinal pigmented epithelium, thus modifying the visual cycle [166,167]. In particular, administered orally, it is able to lower the amount of retinol available in the vision cycle, thus reducing the number of toxic fluorophores that accumulate in the cells of the RPE.

To counteract cell death in AMD, an important and significant role is supported by neuroprotection.

Brimonidine deserves attention as it is a molecule used for several years in ophthalmology for the treatment of glaucoma as it has an alpha-2 agonist action. This molecule is a third generation α2-agonist molecule, a hypotonizing active drug with reduced side effects. In addition to its effectiveness in reducing intraocular pressure, some in vitro and in vivo studies suggest neuroprotective action. The activation of α2-receptors is, in fact, able to modulate the activation of NMDA channels due to excessive increases in glutamate. This consequently reduces intracellular calcium levels which activate the caspase enzymes responsible for the apoptotic process [168].

For the treatment of neurodegenerative diseases, such as amyotrophic lateral sclerosis, ciliary neurotrophic factor (CNTF), identified in the neurons of the ciliary ganglion, is widely studied. CNTF is a cytokine that belongs to the interleukin-6 family; there is a receptor for CNTF located on the membrane of the Muller glial cells and at the level of the retinal photoreceptors. In the ocular field, this compound has already demonstrated effectiveness in slowing the progression of photoreceptor degeneration in animal models of retinitis pigmentosa [169]. The NT-501 is an intraocular implant, in which genetically modified human cells are able to produce CNTF and make it available to the retina. A study demonstrated that NT-501 is well tolerated, produces a significant increase in retinal thickness and total macular volume, and slows visual loss in patients with AMD (ClinicalTrials.gov ID: NCT00447954; https://clinicaltrials.gov/study/NCT00447954, accessed on 4 October 2024).

In addition, the implantation of mesenchymal stem cells, thanks to the therapeutic effects of the secretome produced on the residual cells of the neuroepithelium, allows the control of oxidation to occur through increased metabolic activity, limits apoptotic mechanisms, regulates inflammatory phenomena, and improves neuro-retinal microcirculation [170].

Today, great hopes are placed in gene therapy stopping dry AMD. A new genetic therapy that could stop dry maculopathy is represented by GT005. GT005 is a virus which, injected under the retina, is capable of modifying the genes of pigmented epithelial cells. These cells return to function well, and, therefore, the progression of dry maculopathy is stopped. The treatment aims to restore the balance of an overactive complement system by increasing the production of the Cfi protein. The Cfi protein regulates the activity of the complement system, and it is thought that increasing the production of this protein can reduce inflammation, with the goal of preserving vision. Unlike what happens with wet maculopathy, in which the injections inside the eye must be repeated several times, the new gene therapy for dry maculopathy is administered only once and, if effective, the result would last forever (ClinicalTrials.gov ID: NCT03846193; https://clinicaltrials.gov/study/NCT03846193, accessed on 4 October 2024).

## 8. Drug Repurposing for AMD Treatment

The need for new AMD treatments remains high. The development of new drugs is difficult because of high failure rates and exorbitant costs. With this in mind, clinicians and researchers have been focused on more efficient and economically sustainable alternatives for AMD therapy, including drug repurposing. This strategy implies the use of already approved drugs, known to be safe to use, for other medical conditions. In the AMD context, drugs such as metformin, statins, levodopa, entacapone, fluoxetine, idebenone, and dymetilfumarate have been investigated.

Metformin is a commonly used drug for the treatment of type II diabetes due to its ability to inhibit hepatic gluconeogenesis and increase insulin sensitivity by increasing peripheral glucose uptake and utilization. Interest in its therapeutic potential in the context of AMD stems from its ability to influence age-related metabolic and cellular processes, including inflammation, oxidative stress, and autophagy, all of which are implicated in the pathogenesis of AMD [171]. The administration of metformin to murine models of retinal degeneration has demonstrated that this drug can cross the blood–retinal barrier and stimulate AMPK, thereby protecting photoreceptors from light damage and enhancing the resistance of RPE cells to oxidative stress [172]. Several population-based studies suggested a correlation between the use of metformin and reduced risk of developing AMD. For example, Blitzer et al. reported that metformin use was associated with a reduced risk of developing AMD, especially at low and moderate doses [173]. In another retrospective cohort study of subjects with type 2 diabetes, Chen et al. found that longer treatment with metformin and a higher average daily dose were both associated with a lower risk of AMD [174]. However, other studies have found no such correlation. For example, a nested case–control study by Lee et al. found no association between metformin use and reduced risk of developing AMD in the study population [175]. In addition, a retrospective cohort study conducted by Eton et al. found that current metformin use was associated with an increased risk of dry AMD, whereas previous use was associated with a reduced risk [176]. It is evident that further inquiry is required to gain deeper insights on the protective role of metformin in AMD development.

Statins are HMG-CoA reductase inhibitors, widely used in order to reduce cholesterol levels and prevent cardiovascular diseases. As said before in this review, lipids accumulate in Bruch’s membrane and can interact with ROS and form drusen. In addition, the observation that there are numerous shared risk factors between cardiovascular disease and AMD (e.g., atherosclerosis, cigarette smoking, hypertension, hypercholesterolemia) prompted investigations into the potential use of statins in the management of AMD [171]. However, the results of observational studies investigating the potential association between statin use and the risk of age-related macular degeneration (AMD) have been inconclusive. While some studies have reported a protective effect [177,178,179], others have failed to detect any correlation [180,181,182,183,184], while a third group of studies have suggested that there might be an increased risk [185,186]. These findings are hindered by several limitations. In particular, the lack of specific data on the type and dosage of statins used, the difficulty in controlling for confounding factors, and the variability in study methodologies may have contributed to the inconsistency in results. Furthermore, the majority of observational studies are unable to demonstrate a causal relationship between statin use and AMD. Large-scale randomized controlled clinical trials are needed to confirm or disprove a causal association and to evaluate the efficacy and safety of statins in the treatment of AMD. Sources also suggest that the concentration of statins in the retina could be an important factor in their efficacy in the treatment of AMD. Moir et al. showed that simvastatin reaches a higher concentration in the retina than atorvastatin and pravastatin, while rosuvastatin appears not to cross the blood–retinal barrier [171].

Levodopa (L-DOPA) is employed in the management of the symptoms associated with parkinsonism, particularly in the context of Parkinson’s disease, post-encephalitic parkinsonism, and symptomatic parkinsonism resulting from carbon monoxide or manganese intoxication. L-DOPA is converted to dopamine in the brain, thereby compensating for the dopamine deficiency that occurs in Parkinson’s disease. It has been in use for over half a century and is regarded as a safe and cost-effective treatment. It is hypothesized that L-DOPA is able to interact with the GPR143 receptor, expressed in RPE cells. Activation of this receptor has been observed to increase the production of PEDF (pigmented epithelial-derived factor), a potent antiangiogenic factor, and to reduce the production of VEGF, a strong pro-angiogenic factor whose overexpression is considered one of the main causes of neovascular AMD. Modulating PEDF and VEGF, L-DOPA may exert an influence on angiogenesis and inflammation, which are pivotal processes in the development and progression of AMD. An environment characterized by elevated levels of PEDF and diminished levels of VEGF may hinder the formation of aberrant new blood vessels in the neovascular form of AMD, and mitigate the chronic inflammation that contributes to the degeneration of RPE cells in the dry form [187,188,189]. Several observational studies have demonstrated a correlation between L-DOPA intake and a reduced incidence of AMD, including both its dry and neovascular forms. In a study published in 2016, Brilliant et al. found that patients with a history of L-DOPA prescription had a significantly higher mean age of onset of AMD than those without a prescription. The protective effect was observed in both forms of AMD, including the neovascular form [187]. A review of the Vestrum Health Retina database revealed that patients with neovascular AMD who were exposed to L-DOPA required fewer intravitreal injections over a two-year period, which suggests a potential impact on disease progression. Furthermore, the data indicated that L-DOPA exposure was associated with a reduced risk of conversion from non-neovascular to neovascular AMD over a five-year period [189]. The analysis of the Merative MarketScan Database demonstrated that L-DOPA use was associated with a 15% reduction in the odds of developing neovascular AMD, even after accounting for other risk factors. Additionally, the results indicated a dose–response effect, with doses of L-DOPA between 100 and 300 mg per day for two years associated with a 15% reduction in risk and doses above 300 mg per day associated with a 23% reduction [189]. Further research is required in the form of randomized, controlled clinical trials to confirm the efficacy of L-DOPA in the treatment of AMD and to determine the optimal dosage and duration of treatment. It is important to note that the majority of studies conducted up to this point have involved patients undergoing treatment for Parkinson’s disease with L-DOPA. Consequently, further studies on AMD are required to ascertain whether the observed effects are attributable to L-DOPA alone, or whether they are influenced by other factors associated with Parkinson’s disease. In addition to oral administration, topical administration of L-DOPA may offer a promising avenue for exploration, given its ability to cross the blood–retinal barrier [189].

Entacapone, a pharmaceutical agent that has demonstrated efficacy in enhancing the effects of L-DOPA in the management of the symptoms of Parkinson’s disease, has been identified as a potent inhibitor of the aggregation and fluorescence of N-retinylidene-N-retinylethanolamine (A2E), a lipofuscin component that has been linked to oxidative stress in dry AMD. Mechanistic studies suggest that entacapone redirects A2E aggregation to non-toxic oligomers. In this manner, entacapone diminishes the production of ROS in RPE cells subjected to blue light, thereby preserving them from phototoxic injury [190].

Another drug that has been studied as potentially reducing AMD progression is fluoxetine, a selective serotonin reuptake inhibitor (SSRI) commonly used to treat clinical depression. Ambati et al. conducted an analysis of two distinct health databases, Truven and PearlDiver Mariner, and found that patients exposed to fluoxetine exhibited a significantly lower likelihood of developing dry AMD compared to the control group. The potential mechanism of action against AMD of fluoxetine does not appear to be directly related to its SSRI activity; indeed, it may act by inhibiting the NLRP3 inflammasome, a multiprotein complex involved in innate immunity and inflammation, whose activation leads to the production of pro-inflammatory cytokines, such as IL-1β and IL-18. In vitro studies have shown that fluoxetine can bind to NLRP3 and inhibit inflammasome assembly and activation, and in a murine model of Alu RNA-induced RPE degeneration, it protected RPE cells from degeneration, while other antidepressants tested did not show the same effect [191]. Nevertheless, the role of NLRP3 in AMD remains a topic of contention, and further research is necessary to substantiate the efficacy of fluoxetine in the treatment of AMD.

Idebenone is already approved for the treatment of Leber’s hereditary optic neuropathy (LHON) and may have promising applications in other eye diseases, including AMD. It is a coenzyme Q10 analog with antioxidant properties. This molecule serves as a carrier of protons and electrons from complexes I and II to complex III within the mitochondrial respiratory chain. The precise mechanisms through which it exerts its antioxidant and cytoprotective properties remain to be fully elucidated. An in vitro study conducted on the ARPE-19 cell line demonstrated idebenone’s notable capacity to diminish the onset of cellular senescence, characterized by the expression of the enzyme β-galactosidase and an accumulation of ROS within H_2_O_2_-treated RPE cells. Moreover, it reduced the formation of histone-bound DNA fragments in these cells, which may contribute to the preservation of genomic stability. The treatment with H_2_O_2_ resulted in a substantial elevation in BAX levels, a reduction in Bcl-2 levels, and a subsequent augmentation in the BAX/Bcl-2 ratio; the utilization of idebenone led to a notable decline in BAX levels and an increase in Bcl-2, thereby restoring the BAX/Bcl-2 ratio to its normalized state. In conclusion, idebenone increases ARPE-19 cell survival and reduces cell death, senescence, and oxidative stress by stabilizing the BAX/Bcl-2 ratio [192,193]. These in vitro results suggest that idebenone may possess the required properties to be a potential therapeutic agent for AMD.

Dimethylfumarate (DMF) is currently approved and used for the treatment of psoriasis and multiple sclerosis. In addition to the approved indications, DMF is the subject of numerous clinical studies designed to evaluate its efficacy in other diseases characterized by inflammation and/or oxidative stress, including cerebrovascular disorders, autoimmune diseases and cancer. DMF may represent a promising avenue of treatment for AMD, particularly given its capacity to strongly activate the Nrf2 signaling pathway. This pathway plays a crucial role in the maintenance of cellular integrity by protecting cells from oxidative stress and inflammation. Specifically, the DMF mechanism of action involves the inhibition of the Keap1 protein, which is responsible for repressing Nrf2 activity in the cytoplasm. By impeding Keap1, DMF permits Nrf2 to translocate into the nucleus, where it binds to antioxidant response elements (AREs). This interaction results in the activation of the transcription of genes encoding for antioxidant and protective enzymes, thereby enhancing cellular defenses against oxidative stress. DMF can protect RPE cells from oxidative stress increasing glutathione (GSH) levels in human RPE cells, improving their ability to neutralize ROS, and from inflammation reducing the expression of inflammatory cytokines such as IL-6, IL-8, MCP-1, and VEGF, attenuating inflammation at the retinal level [194]. The efficacy of DMF in the treatment of AMD must be confirmed by additional preclinical and clinical studies, which will also determine the optimal dosage and duration of treatment. Moreover, the development of a topical formulation of DMF is a necessity for the treatment of eye diseases: the administration of the drug in this way could enhance its efficacy and reduce the incidence of systemic side effects.

In their study, Wang et al. identified other drugs that may have the potential to reduce the progression of wet AMD. These include megestrol acetate, erlotinib, epoetin alpha, and donepezil. However, it should be noted that the evidence regarding the efficacy of these drugs is currently limited and further research is needed [195].

## 9. Conclusions

This review provides a comprehensive overview of AMD, focusing on both forms and underlining the lack of effective treatments for the dry form. AMD is the main cause of central vision loss among individuals over 50 years. Dry AMD is the most common form of the disease. In this case, waste materials such as lipids, minerals, and proteins deriving from apoptotic cells accumulate, and atrophy of the RPE and the photoreceptor layer occurs. Conversely, wet AMD is characterized by an abnormal expression of VEGF which promotes neoangiogenesis. New blood vessels grow under the macula and have the tendency to leak fluid, leading to hemorrhages and fibrosis.

Figure 1 summarizes the main approaches to treating wet and dry AMD.

The gold-standard therapy for the treatment of wet AMD is represented by intravitreal injections of anti-VEGF drugs, which are able to limit VEGF function, including Ranibizumab^®^, Brolucizumab^®^, and Aflibercept^®^. These agents have been demonstrated to stabilize the progression of the disease, significantly slowing down the progression of blindness. The major disadvantage is represented by the short half-life of the molecules and the consequent need for repeated injections to maintain the efficacy of the therapy. In individuals who do not respond to anti-VEGF therapy, PDT may represent an alternative option for the treatment of wet AMD, aiming to cause vaso-occlusion of the pathologic blood vessels. Recent studies have demonstrated that PDA may lead to compensatory upregulation of angiogenic factors, paving the way for further research concerning the combination of PDT and anti-VEGF agents. Current research is focused on the development of new therapeutic options that provide alternatives to routine injections in order to increase the compliance of patients. One of the most promising approaches includes the improvement of the delivery system of anti-VEGF drugs in order to increase its action. An example is represented by Susvimo^®^, a reservoir that can be implanted in the eye and releases Ranibizumab over a period of 6 months. Gene therapy is another promising alternative to repeated injections of anti-VEGF drugs, since it is a one-time treatment which should make the eye able to produce its own anti-VEGF agents. Nevertheless, the long-term efficacy of gene therapy is still uncertain.

Although significant progress has been made in the treatment of wet AMD, dry AMD remains a significant challenge. Current treatment options for dry AMD are limited to management strategies (e.g., dietary supplementation with antioxidants, vitamins and minerals, quitting smoking, following a balanced diet, exercising on a regular basis), but promising research on several new therapies is ongoing. Over the past few years, several approaches have been investigated in order to reduce the progression rate of the disease by intervening against oxidative stress, chronic inflammation, and toxic derivatives of normal visual function. Among the drugs under study, hydroxylamine OT-551 exerts antioxidant, anti-inflammatory, and anti-angiogenic effects, and the obtained data show a possible effect in visual acuity maintenance. Other drugs under evaluation for their antioxidant properties are Elamipretide, which targets cardiolipin in mitochondria, and Risuteganib. Since dry AMD is characterized by an excessive activation of the complement cascade, drugs able to counteract chronic inflammation are currently under study, such as Pegcetacoplan and POT-4, both targeting complement factor C3. Given that lipofuscin accumulates over years as a consequence of photoreceptor metabolism, limiting rod activity may be a strategy to decrease toxic molecules in the retina. An example of a drug able to modulate visual cycle is Fenretinide, which reduces the quantity of retinol available and, consequently, the toxic metabolites accumulating in the RPE. The last therapeutic approach under evaluation concerns neuroprotective drugs such as Brimonidine, in order to counteract cell death. As in the case of the treatment for wet AMD, hopes are also high for gene therapy in AMD.

In the last few years, a number of promising candidates have been identified that may eventually be repurposed for the treatment of AMD. However, the majority of these have been identified through observational studies, which have limited capacity to demonstrate causality rather than mere correlation. Furthermore, the dosing regimens of these agents has remained underexplored.

In conclusion, dry AMD is a chronic and progressive condition, and even with management, vision loss can occur over time. Currently, research is actively exploring several promising therapeutic approaches.

## Figures and Tables

**Figure 1 ijms-25-13053-f001:**
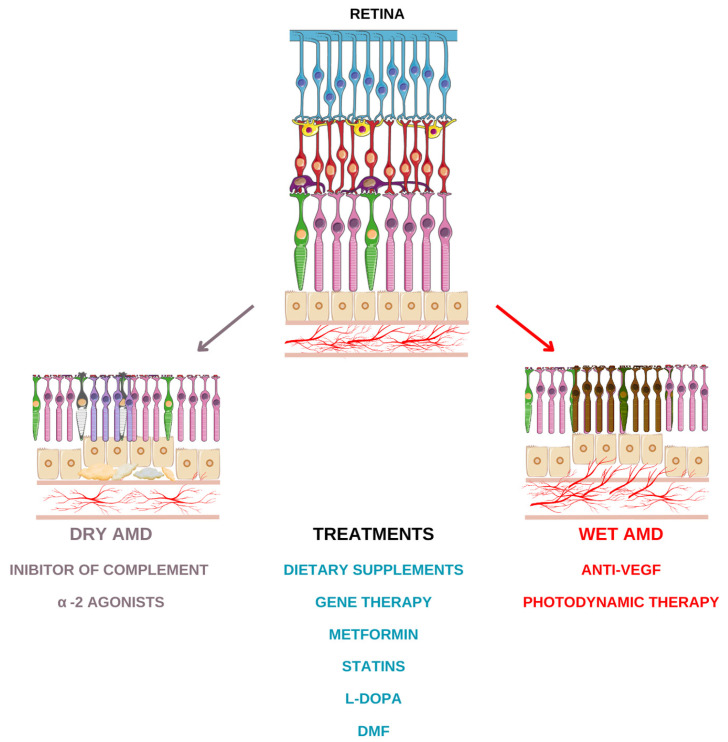
Main AMD treatments. The purple section illustrates the treatments for dry AMD, whereas the red section depicts the treatments for wet AMD. The treatments that are not related to either form are summarized in the central area in blue. This figure has been created by the authors using Canva and Smart—Servier Medical Art.

**Table 2 ijms-25-13053-t002:** Summary of FDA-approved anti-VEGF drugs.

Name	Commercial Name/Sponsir	Type/Approved Year	Mechanism of Action	Dosage
Bevacizumab	Avastin Genentech	mAb Off label	VEGF-A inhibition	1.25 mg/0.05 mL
Pegaptanib	Macugen Bausch/Lomb	aptamer 2004	VEGFA165 inibhition	0.3 mg/0.09 mL
Ranibizumab	Lucentis Genentech	Fab 2006	VEGFA inhibition	0.5 mg/0.05 mL
Aflibercept	Eylea Regeneron	FcFP 2011	VEGF-A, VEGF-B, and PGF inhibition	2 mg/0.05 mL
Conbercept	Lumitin Cheongdu Kanghong Biotech	FcFP 2013 (China)	VEGF-A, VEGF-B, and PGF inhibition	0.5 mg/0.05 mL
Brolucizumab	Beovu Norvatis	scFv 2019	VEGF-A inhibition	6 mg/0.05 mL
Ranibizumab	Susvimo Genentech	mAb implant 2021	VEGF-A inhibition	2 mg/0.02 mL, (6 months)
Faricimab	Vabysmo Genentech	mAb 2022	VEGF-A and Ang2 inhibition	6 mg/0.05 mL

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
