# Peer review of "Different Therapeutic Approaches for Dry and Wet AMD"

_ijms, 2024, doi:10.3390/ijms252313053_

Round 1
Reviewer 1 Report
Comments and Suggestions for Authors
The review titled” Different therapeutic approaches for dry and wet AMD” by Marchesi N. et al. concerns the therapeutic approaches currently in use for the dry and wet forms of AMD. The topic is not new and has already been reported several times in other reviews also mentioned by the authors and the therapies currently in use have not even been all reported in this review. Just to make a mention, there is an extensive literature of clinical trials about the effects of saffron on patients with AMD, completely ignored by the authors and that would deserve to be included as therapies based on natural molecules (doi: 10.1136/bmjophth-2023-001399; doi: 10.4103/1673-5374.274325; doi: 10.3390/nu11030649; doi: 10.1007/s00417-018-4163-x. Epub 2018; PMID: 28289690; doi: 10.1186/1479-5876-11-228; doi: 10.1155/2012/429124).
Furthermore, preclinical and clinical research is making great efforts to develop curative therapies that should be mentioned. In fact, starting from the assumption that neurodegenerative diseases often share similar causes and dysfunctional mechanisms, recently there has been investigation into whether drugs already in use for other disorders can also have a therapeutic effect for AMD. One of these is L-DOPA, a gold standard drug for Parkinson's disease, which could be repositioned in the treatment of AMD (doi: 10.1016/j.amjmed.2020.05.038; doi: 10.1016/j.amjmed.2015.10.015; doi: 10.1016/j.oret.2023.04.014).
Therefore, to make the review original and up to date the authors should improve the contents, mostly highlighting the main ongoing clinical trials.
In this version the review is not publishable.
Reviewer 2 Report
Comments and Suggestions for Authors
Different therapeutic approaches for dry and wet AMD are important to be discussed and summarized. Therefore, this manuscript is interesting. Several comments should be addressed.
1) Wet AMD vs Dry AMD; their ratios should be presented with clear numbers.
2) Wet AMD vs Dry AMD; mimicking experimental models in vitro and in vivo could be listed.
3) In Wet AMD section, how about subretinal fibrosis? This aspect is not well discussed. Expanding this part is needed.
4) In Dry AMD section, photoreceptor damage with molecular biological points is not well discussed. Expanding this part is also needed.
5) Complications of AMD with other ocular or metabolic diseases (diabetes, hyperlipidemia, or ETC) should be well-covered.
Round 2
Reviewer 1 Report
Comments and Suggestions for Authors
I appreciated the authors’ efforts in addressing my previous comments. The revisions have significantly improved the contents and the quality of the review.
I also appreciate that the authors added a paragraph on experimental models to study AMD, probably suggested by the other referee. In this regard, however, the authors completely omitted the AMD model based on light damage, which reproduces many of the characteristics of both dry and wet AMD. Therefore, it is necessary to mention this model. I suggest to the authors a recent review (doi: 10.1515/revneuro-2023-0130)
Reviewer 2 Report
Comments and Suggestions for Authors
The raised comments are all addressed. There is no further main comment.
A minor "optional" suggestion is to draw one illustration to summarize the current manuscript with current and future drugs to cure dry and wet AMD. This is because the current work contains long contexts without illustrations.
But it does not mean it should contain this image.
